# Neuroinflammation: Integrated Nervous Tissue Response through Intercellular Interactions at the “Whole System” Scale

**DOI:** 10.3390/cells10051195

**Published:** 2021-05-13

**Authors:** Daniele Nosi, Daniele Lana, Maria Grazia Giovannini, Giovanni Delfino, Sandra Zecchi-Orlandini

**Affiliations:** 1Section of Histology anf Human Anatomy, Department of Experimental and Clinical Medicine, University of Florence, Largo Brambilla, 3, 50134 Florence, Italy; sandra.zecchi@unifi.it; 2Section of Clinical Pharmacology and Oncology, Department of Health Sciences, University of Florence, Viale Gaetano Pieraccini, 50139 Florence, Italy; daniele.lana@unifi.it (D.L.); mariagrazia.giovannini@unifi.it (M.G.G.); 3Department of Biology, University of Florence, Via Madonna del Piano, 6, 50019 Sesto Fiorentino, Florence, Italy; giovanni.delfino@unifi.it

**Keywords:** inflammation, cell–cell interactions, aging, neurodegenerative diseases, extracellular matrix

## Abstract

Different cell populations in the nervous tissue establish numerous, heterotypic interactions and perform specific, frequently intersecting activities devoted to the maintenance of homeostasis. Microglia and astrocytes, respectively the immune and the “housekeeper” cells of nervous tissue, play a key role in neurodegenerative diseases. Alterations of tissue homeostasis trigger neuroinflammation, a collective dynamic response of glial cells. Reactive astrocytes and microglia express various functional phenotypes, ranging from anti-inflammatory to pro-inflammatory. Chronic neuroinflammation is characterized by a gradual shift of astroglial and microglial phenotypes from anti-inflammatory to pro-inflammatory, switching their activities from cytoprotective to cytotoxic. In this scenario, the different cell populations reciprocally modulate their phenotypes through intense, reverberating signaling. Current evidence suggests that heterotypic interactions are links in an intricate network of mutual influences and interdependencies connecting all cell types in the nervous system. In this view, activation, modulation, as well as outcomes of neuroinflammation, should be ascribed to the nervous tissue as a whole. While the need remains of identifying further links in this network, a step back to rethink our view of neuroinflammation in the light of the “whole system” scale, could help us to understand some of its most controversial and puzzling features.

## 1. Introduction

In the central nervous system (CNS), a basic classification of the different cell populations can be formulated, according to their specific activities under normal and pathological conditions. Neurons (nerve cells) are the morpho-functional units involved in collecting and integrating external and internal stimuli, elaborating them in accordance with memorized experience and orchestrating the responses of the organism. Non-neuronal (glial) cells concur to the maintenance of adequate physiological conditions (homeostasis) in the nervous tissue micro-environment. Among glial cells, astrocytes are known to perform “housekeeping” activities in the CNS as they contribute to the homeostasis of neuronal networks, regulate nerve cell maturation and modulate neurotransmission [1]. Microglia represent the immunocompetent cells of the CNS [2] and have been shown, in addition, to perform several maintenance activities in the normal nervous tissue [2,3,4,5], which may be synergic with those of astrocytes. For example, both cell types remove exceeding amounts of neurotransmitters at synaptic sites and modulate connectivity of the neuronal network [1]. Therefore, it is not surprising that several heterotypic interactions between these two cell populations, occurring under both physiological and pathological conditions, have been reported in the past few years. These heterotypic interactions, which become extremely intense during neuroinflammation, have been widely studied in the field of neurodegenerative diseases. It is known that the onset of age-related neurodegeneration, such as in Parkinson’s (PD), Huntington’s (HD) and Alzheimer’s (AD) diseases, leads to the activation of the inflammatory response in the CNS. Denatured amyloid peptides and nerve cell debris accumulating in the nervous tissue trigger the production of pro-inflammatory cytokines by neurons and astrocytes [6,7] which, in turn, shift the functional activity of microglia from a surveillance/maintenance mode into active phagocytosis [2,8,9]. On the one hand, microglia perform phagocytosis of Aβ-deposits during neuroinflammation, thus contributing to the clearance of amyloid peptides and cytotoxic debris from the brain [2,10,11,12,13,14]. On the other hand, prolonged microglia activity may exacerbate neuroinflammation and, in turn, increase the production of amyloid fibrils, thus intensifying neurodegeneration [15,16]. Moreover, microglial phagocytosis of living, healthy neurons has also been reported in the inflamed CNS [4,11,13,14,17]. The functional shift, from a neuroprotective to a neurodegenerative activity, has been associated with different phenotypes of microglia, which may range from anti-inflammatory to pro-inflammatory, respectively. A further element of complexity in this scenario relies on some peculiarities of aging microglia, which is characterized by anomalies in distribution, morphology and phagocytic marker expression, as well as low efficiency in the clearance of pro-inflammatory molecules [18,19,20,21,22,23]. Conversely, the identification and characterization of microglial phenotypes in neuroinflammation led to the development of several therapeutic strategies against neurodegenerative diseases [24,25], which aim at reverting the neurodegenerative polarization of microglia. Undoubtedly, these therapeutic approaches could be implemented by the knowledge of the environmental factors that trigger such polarization.

Astrocytes are synergic players with microglia in the neuroinflammatory response, directly performing clearance of amyloid species from the CNS [26,27] and influencing microglial phagocytosis [1,2,28,29,30,31,32,33,34]. Nevertheless, it has been suggested that in chronic neuroinflammation the phenotype of astrocytes may also vary from anti-inflammatory, neuroprotective [35,36,37] to a pro-inflammatory neurodegenerative phenotype [35,36,37], which should be induced by pro-inflammatory reactive microglia [35,36]. Therefore, the inhibition of the pro-inflammatory phenotype of microglia is expected to also hinder the expression of the pro-inflammatory phenotype of astroglia. On the other hand, if we assume that the ubiquitous meshwork of astrocyte projections may be the first glial structure making physical contact with pro-inflammatory molecules in the nervous tissue, the effects of such cytotoxic interactions on astrocytes and on their activities should be carefully pondered. These considerations hint that the identification of a reliable sequence of mutual inductions and interdependencies linking astrocytes and microglia could be deduced by integrating their patterns of inflammatory activation and phenotype polarization. Increasing knowledge of such sequential relationships is leading to relevant advances in the field of neurodegenerative diseases. However, this task is complicated by the intense molecular crosstalk linking these glial cells and the occurrence of close mechanical interactions that also plays a relevant role in these processes [33]. Accordingly, this review, rather than providing a description of astroglia–microglia interactions during inflammation, will attempt to contextualize these reciprocal influences, with the aim of discussing how they might affect, or be affected by, the whole system. We believe that this approach could be fruitfully adopted to conceive new diriment questions and highlight unexpected features of the unsolved enigma of neuroinflammation.

## 2. Neuroglia

The existence of neuroglia was first hypothesized by Rudolf Virchow, although what he described did not exactly correspond, in its morphological-functional traits, to the modern glial tissue concept. He opposed the leading idea that nervous tissue has no connective-like support, and in 1856 he wrote, “this connective substance forms…a sort of putty (neuroglia), in which the nervous elements are embedded” (quoted in [38]). Otto Deiters in 1865 drew the first image of stellate cells lacking axon in the CNS (shown in [38]). Later, Golgi took advantage of his innovative silver staining technique and definitely confirmed the intuition of Dieters, thus paving the way to the modern studies on glial cells [38]. The outcomes of these investigations allowed to overwhelm the early view of neuroglia as a mere “neural glue” by providing evidence of a manifold supporting role toward neurons, which involves a variety of glial cell types in CNS, PNS and sense organs.

## 3. Microglia

### 3.1. Microglia in Health

The terms “microglia” and “oligodendroglia” were introduced in 1919 by Pío Del Río-Hortega, who described them as components of the “Third Element” of the cell repertory in the nervous tissue, to be recognized as different from nerve cells and already known lines of gliocytes [39]. Microglia represent the myeloid cells of the CNS and derive from yolk sac erythro-myeloid progenitors entering the nervous tissue in the early phases of CNS development, before completion of the blood-brain barrier. In these phases, both migration and positioning of embryonic microglia are orchestrated by neuronal progenitors, secreting C-X-C motif chemokine 12 (Cxcl12), a ligand of microglial C-X-C chemokine receptor type 4 (CxcR4) [40]. They differ from peripheral macrophages in terms of protein expression. Microglia are characterized by specific markers such as transmembrane protein 119 (TMEM119), P2Y purinoceptor 12 (P2RY12) and Sal-like protein 1 (SALL1) [41]. Moreover, they express low levels of cluster of differentiation (CD45) protein and major histocompatibility complex (MHC) II molecules, which are highly represented in non-parenchymal macrophages [41]. Cell transplantation experiments in mouse models demonstrated that this phenotype signature is due to both environment conditioning and differences in their adaptive potential to the CNS environment with respect to peripheral macrophages [42]. Within the CNS, differential gene expression of microglia in white and gray matter has been correlated with different environmental conditions [43]. In white matter microglia, the inhibitor genes of the nuclear factor kappa-light-chain-enhancer of activated B cells (NF-κB) pathway were more expressed than in gray matter microglia. Conversely, microglia showed higher expression of genes involved in the interferon (IFN) type I response in the gray matter than in the white matter. Under physiological conditions, microglia are localized according to a regular pattern [44,45] and show a ramified shape, several branched processes extend from the soma of these cells (Figure 1A) and act as environmental sensors. Microglia have been recently demonstrated to undergo cyclic turnover [45,46,47] and, even after acute ablation, they are able to restore a proper homeostatic density, through mechanical cell–cell signaling [45]. Activities of microglia in normal nervous tissue range from the regulation of neurogenesis [48], synaptic density [49], connectivity [50] and plasticity [51] to the prevention of neurotransmitter cytotoxicity that follows exceeding release from synapses [52,53]. These activities are modulated via numerous interactions between microglial surface receptors and the extracellular matrix (ECM) and surface or soluble factors from other cell lineages, respectively. For example, it has been demonstrated that synapse plasticity is promoted by microglia through local remodeling of the extracellular matrix (ECM) [51]. Neurons may direct such activity either by releasing the cytokine interleukin 33 (IL-33) [51] or through intercellular contacts [54]. From a general perspective, the mosaic pattern of distribution in the nervous tissue, and the ramified cell morphology of microglia, may favor their activities, as well as chemical and mechanical interactions with astrocytes and neurons.

### 3.2. Microglia in Neuroinflammation

The “active” state of microglia during neuroinflammation is known to be rapidly induced by alterations of the physiological homeostasis in the nervous tissue. During neuroinflammation, microglia have been labeled as M1-neurotoxic or M2-neuroprotective, according to categories characterized by different patterns of activation and production of cytokines [56]. However, reasonable doubts were well expressed by Ransohoff [57] in 2016 regarding the very existence of M1 and M2. In our opinion, although suitable for summarizing functional differences in microglial activities, M1 and M2 can be considered, at most, two conceptual opposites in a wide range of activation states that microglia can assume during neuroinflammation. Indeed, recent transcriptome studies in different models of neurodegeneration revealed that “activated” microglia may simultaneously express both neurodegenerative and neuroprotective factors [58]. Furthermore, transcriptome studies on murine models of AD, supported by microscopy analyses on human post-mortem brains, identified a neuroprotective subset of disease-associated microglia (DAM), whose full activation depends on a transmembrane receptor protein that controls microglial activity and survival: the triggering receptor expressed on myeloid cells 2 (TREM2) [59]. An additional element of complexity, which should be carefully considered in the design of microglial experiments, is provided by a marked sex-specific diversification of microglia, affecting both time course and intensity of their reactivity [60].

Currently, we know that the transition of microglia activity from neuroprotective to neurodegenerative is dependent upon time. In fact, whenever the physiological homeostasis of the CNS is restored, microglia return to their “resting” state [45]. On the contrary, when the activity of these cells is prolonged, overproduction of proinflammatory cytokines may exacerbate neuroinflammation and promote neurodegenerative effects. The main function played by activated microglia is the phagocytosis of invading viruses [61], bacteria and potentially neurotoxic molecules, ranging from misfolded peptides to cell debris and whole apoptotic neurons. According to the main functional path of active macrophages, this clearance process can be summarized by three sequential steps: (i) the “find me” step that involves migration of microglia towards the target; (ii) the “eat me” step that refers to the phagocytic process; (iii) the “digest me” step that consists of the degradation of the engulfed cargo. Dysregulation of these sequential mechanisms may produce neurodegenerative effects during chronic inflammation. For example, conditions of stress induced by chronic inflammation may cause an erroneous exposition of molecules, such as phosphatidylserine and desialylated glycoproteins on plasmamembranes of viable neurons. Eventually, these nerve cells become unwanted targets of phagocytosis, since the extracellular moieties of the exposed molecules were found to act as “eat me” signals for microglia in the human epileptic brain [62], in lipopolysaccharide (LPS) treated primary microglial and neuronal co-cultures [63], as well as in murine models of neuroinflammation induced by LPS [64] and endothelin-1 [65] injection. Also, the Ca^2+^-binding chaperone protein calreticulin has been demonstrated to act as an “eat me” signal when exposed to the neuron’s plasma membrane in cell-culture models of LPS and Aβ induced neuroinflammation [66]. It appears, therefore, that phagocytosis of healthy neurons results from dysregulation of their enzymatic mechanisms, rather than real microglial dysfunction. Conversely, dysregulation of the “find me” mechanisms could be one of the possible impairments of phagocytosis entailing neurodegenerative effects, imputable to microglia. It is known that neurodegenerative microglia exhibit unbranched ameboid shape [67], suggesting a low efficiency of their targeting mechanisms. Indeed, microglial targeting in the “find me” step involves dynamic modifications of their cytoplasmic processes (Figure 1B), which show increased branching towards target molecules, based on chemical as well as mechanical stimuli [2,33]. Accordingly, it has been demonstrated that in an aged brain, senescent microglia show decreased branching (Figure 1C) and migration rate [68]. Moreover, our studies in the hippocampus of aged rats revealed that the unbranched amoeboid morphology of microglia was correlated with defective targeting and clearance of proinflammatory molecules [33].

A mainly unanswered question concerning microglia in inflammation relies on the reasons for their shift from a neuroprotective to a neurodegenerative phenotype. Apparently, the nervous tissue environment is an important definer of microglia identity, and it may also induce activation of microglial signature genes in myeloid cells derived from different stem compartments [42]. Indeed, in human multiple sclerosis (MS) brains, white matter and gray matter microglia showed environment-specific transcriptional differences [43]. Differentially expressed genes were associated with lipid metabolism, lysosomal function and foam cell formation in white matter microglia and with glycolysis and iron homeostasis in gray matter microglia. Interestingly, the total number of differentially expressed genes in white and gray matter was lower in MS than in the normal brain, thus confirming environmental conditioning as a main determinant of microglia diversity. Indeed, along with regional and pathology-specific effects, altered homeostasis, as well as neuroinflammation itself, pose common threads to microglia. For example, the activation of inflammatory signaling inherently promotes depolarization of mitochondrial membranes and alarmin release in microglia, thus impairing their energy availability and exacerbating their immune response [69,70]. Mitochondrial dysfunction inevitably affects microglia as well as other cell lineages in neurodegenerative diseases such as neurons in AD [70,71]. However, several data suggest that promoting selective microglial autophagy of damaged mitochondria (mitophagy) restores functions of microglia and ameliorates symptoms of AD [70,71]. In this view, further studies aimed at identifying mechanisms of microglia modulation by the CNS environment may be necessary to address the inductive factors of microglial phenotype shift in neuroinflammation.

The CNS can be considered the most regulated environment of vertebrates. A neurovascular unit, composed of endothelial cells of peripheral capillaries along with surrounding pericytes, astrocytes and macrophages, regulates exchanges between blood and neural tissue and contributes to the maintenance of its homeostasis, thereby establishing a sort of tissular ecosystem. In view of this up-to-date blood-brain barrier concept, environmental factors affecting local activities of microglia in the healthy CNS may be associated with their interactions with surrounding cells and/or extracellular matrix. Indeed, intercellular adhesion molecule 5 (ICAM5) shed by neurons was found to promote an anti-inflammatory response from LPS treated microglia in cell culture experiments [72]. Furthermore, intercellular contacts between neurons and microglia in ischemic mouse models were suggested to be involved in synapse remodeling [54]. The effects of mechanical alterations of the ECM on functional activities of nerve cells should not be underestimated, since they may play an important role in neurodegenerative pathologies such as traumatic brain injury [73]. In particular, in vitro studies demonstrated that stiffness of the extracellular substrate and branching of microglia processes are positively correlated [74]. Conversely, in the neocortex of AD mice models, enzymatic degradation of chondroitin sulfate proteoglycans attenuated the ECM, thus promoting microglial activation and Aβ clearance [75]. Further investigations revealed that mechanical stimulation of these cells up-regulates their expression of integrin-β1 [76], a known plasma-membrane mechano-receptor [77]. In detail, binding of microglial integrin-β1 with components of the ECM, such as the mechano-signaling-associated proteins fibronectin and vitronectin, was found to promote microglial reactivity in mouse models of experimental autoimmune encephalomyelitis (EAE), the most used model of MS [78]. Of note, transient expression of fibronectin by microglia was involved in the remyelination process in rat models of toxin-induced demyelination [79]. Accordingly, neonatal microglia, expressing fibronectin and peptidase inhibitors, have been demonstrated to play a relevant role in scar-free spinal cord regeneration after crush injury [80]. On the other hand, the protracted autoimmune activation in EAE rat models, promoted fibronectin aggregation, thus impairing the remyelination process [79]. A further aspect of environment microglia interactions in neuroinflammation involves the loss of integrity and functionality of the neurovascular unit that, besides promoting infiltration of peripheral macrophages, leads to edema formation and modifies tissue osmolarity. In mouse retina models, a hypotonic environment induced microglial swelling and reduced branching, which resulted in the inactivation of the Transient Receptor Potential Vanilloid-4 (TRPV4) channels [81]. Interestingly, the association between TRPV4 and integrin-β1 has been demonstrated to play a relevant role in the transduction of local mechano-chemical stimuli in endothelial cells [82].

In the aged brain, the distribution pattern of microglia is significantly deteriorated and shows a severe reduction of branching [33,44], which is consistent with the immunosuppression evoked by a chronic low-grade inflammation developed with aging (inflammaging) [83]. It has been suggested that a feed-forward relationship between immunosuppression and inflammaging may play a relevant role in age-related diseases [84]. Of note, in the aged nervous tissue, microglia express high levels of phagocytosis markers such as CD68 [19,85]. However, although being able to perform a phagocytic activity, these cells showed inefficient clearance of amyloid deposits in AD [86,87]. It is reasonable to assume that a better understanding of the processes underlying altered microglial distribution and/or migration in the aging brain might lead to a significant broadening of the current knowledge on amyloid diseases.

## 4. Astrocytes

### 4.1. Astrocytes in Health

A general classification of astrocytes in the healthy nervous tissue features two distinct categories: “protoplasmic” astrocytes, which populate the gray matter, representing the most abundant cell population in this CNS component; and “fibrous” astrocytes, which have been detected mainly in the white matter and are characterized by long and straight processes, connecting with blood vessels [88]. Under physiological conditions, the distribution of somata of protoplasmic astrocytes follows a regular pattern, occupying distinct anatomical domains [89,90]. Numerous, highly ramified, cytoplasmic processes extend from these cells throughout the gray matter and create a dense meshwork (Figure 1D). By establishing numerous GAP junctions between their processes, astrocytes may behave as an electrophysiological unit [91], that is, a functional syncytium. This meshwork allows astrocytes to perform maintenance activities in the nervous tissue, such as clearance of glutamate [92,93] and free radicals [94] and buffering of [K^+^] and pH [95].

Current evidence suggests that astrocytes represent a variegate population of cells sharing several properties, possibly displaying specific functional traits in different micro-environments. The cerebral cortex, which is one of the most targeted regions in studies of glial heterogeneity, hosts astrocytes derived from three different sources: radial glia in the embryonic ventricular zone, progenitor cells in the subventricular zone and glial-restricted progenitor cells [96]. In the cell layers of the neocortex of mice, astrocytes exhibit different features, both morphological and transcriptional [97,98]. Moreover, recent studies have identified five subsets of astrocytes, each expressing different markers, that coexist with different frequencies, in the olfactory bulb, neocortex, thalamus, cerebellum, brainstem and spinal cord of mice [99]. Taking into consideration the broad spectrum of astrocyte subpopulations showing molecular, morphological and functional differences, questions arise regarding the mechanism underlying such diversification. Evidence on this subject suggests that environmental factors and heterotypic cell interactions may hold a major part of the answer. In a recent study, Farmer and colleagues [100] showed that, under normal conditions, the diverse specialization fates of astrocytes are significantly affected by their interactions with neurons. Indeed, nerve cells release an array of signals, which drive the complexity of astrocytes during development and pattern their heterogeneity fitting the needs of local neural circuits. Since neuronal population may be regarded here as an environmental factor of astrocytes, the functional syncytium of the latter should be considered, reciprocally, a relevant component of the environment of neurons. Astrocyte interactions with neurons modulate synaptic functions by releasing either glutamine, a precursor of glutamate and gamma(γ)-aminobutyric acid (GABA) [101] or the N-methyl-D-aspartate receptor (NMDAR) coagonist D-serine [102] and by modulating the concentration of neurotransmitters in the synaptic cleft [92,103]. Astrocytes may also modulate synaptic plasticity in developing [50] as well as adult [104] brains and provide metabolic support to neurons [105]. Moreover, two proteins released by astrocytes: HEVIN (High Endothelial Venule Protein) and SPARC (Secreted Protein Acidic and Rich in Cysteine), may enhance or inhibit synaptogenesis, respectively [106]. There is also evidence that astrocytes possibly trigger mechanisms of mechano-signaling to modulate cytoskeletal rearrangement in neurons: they may promote neurite regeneration by producing fibronectin [107] and inhibit neurite outgrowth via direct cell–cell contacts [108]. As a direct contribution to the environmental substratum, astrocytes and their neural progenitors as well produce proteoglycans of the ECM in the developed and healthy brain [109]. Furthermore, the meshwork arrangement of astrocytes may favor their heterotypic interactions with other cell types in the nervous tissue. Astrocyte processes reaching blood vessels release the constitutive vasodilating enzyme cyclooxygenase-1 (COX-1), thus modulating local blood flow in response to changes in neuronal activity [110].

### 4.2. Astrocytes in Neuroinflammation

Recently, Escartin and colleagues [111] suggested that the nomenclature describing astrocyte responses to neurodegenerative events occurring in the CNS, actually features too many terms, such as “astrocytosis”, “astrogliosis”, “reactive gliosis”, “astrocyte activation”, “astrocyte reactivity”, “astrocyte re-activation” and “astrocyte reaction”. In order to solve this redundancy, the authors reduced the plethora of names only to “astrocyte reactivity” while suggesting a new term: “reactive astrogliosis” as a possible and equivalent alternative. Indeed, both terms define the engagement of these cells in response to pathology but the former underlines the competence in taking on specific roles to contrast diverse diseases. This lexical revision reflects an objective difficulty in summarizing cell response that, according to numerous morphological, transcriptomic and genomic studies, has been depicted as a heterogeneous process encompassing a wide range of “cellular, molecular and functional changes” [112].

It has been shown that during neuroinflammation, part of the protoplasmic astrocytes may undergo proliferative activity [113] that leads to the identification of two cellular subsets. Non-proliferative astrocytes retain their original position as well as the volume domains of their processes [114], whereas new proliferating astrocytes are involved in the formation of the “glial scar” [115]. In this process, astrocytes surround and isolate a damaged part of the nervous tissue and perform relevant local roles in preventing leukocyte infiltration and restoring the blood-brain barrier [116]. Moreover, a transcriptomic study on rat models of MS indicated that reactive astrocytes, similar to microglia, may show two opposing phenotypes [36]. The authors identified a neurotoxic, pro-inflammatory A1 and a neuroprotective, anti-inflammatory A2 phenotype, which are characterized by different patterns of activation and gene expression. However, these two phenotypes, as in the M1/M2 paradigm of microglia, should be considered a result of a useful synthesis for illustrating a broad range of different responses that astrocytes may perform in a dynamic environment such as the inflamed nervous tissue. Even the identification of reactive astrocytes may be problematic. A known distinctive feature available to discriminate the onset of the reactive response of astrocytes to several CNS pathologies, consists of marked hypertrophy of their cytoplasmic processes, along with the increase of a cytoskeletal protein, namely the glial fibrillary acidic protein (GFAP) [117]. However, it has been demonstrated that GFAP increase may depend on astrocyte proximity to the injury site [118]. Moreover, our studies in the CA1 hippocampus of aged rat models revealed that sub chronic inflammation may be characterized by a relevant decrease of GFAP expression by astrocytes [55]. This contradictory finding is to be expected when considering the need of contextualizing the inflammatory responses of astrocytes, in order to grasp the mechanisms underlying their heterogeneity [112]. In this view, changes in the mechanical properties of the extracellular matrix, along with heterotypic interactions, may affect astrocyte response to tissue damage. Recent evidence suggests that these cells act as baroreceptors sensing brain perfusion pressure and, in turn, modulating ortho-sympathetic nervous system activity on heart rate and blood pressure [119]. Intense mechanical pressure may per se activate signaling pathways typical of reactive astrocytes [120]. Astrocytes cultured in vitro on different hydrogel substrata of increasing stiffness, showed an increasing spread of processes [121]. Oddly, atomic force measurements disclosed that glial scars are softer than healthy tissue and, therefore, that stiffness of the extracellular matrix and expression of GFAP by astrocytes are inversely correlated [122].

Although a study on the retinal ganglion layer of rats suggests that astrocytes may release an undefined necrotic factor [36], there are reasonable doubts that these cells may acquire specific neurodegenerative functions in inflammation. Rather, it seems that neurodegenerative effects may be related to the loss or impairment of their constitutive functions. In particular, reactive astrocytes showed impairment of [K^+^] buffering in HD [123], glutamate uptake in AD and amyotrophic lateral sclerosis [124,125], GABA modulation in HD and PD [126,127] and weakening of energetic support to neurons in MS [128]. In mouse models of MS, excessive releases of Vascular Endothelial Growth Factor-A (VEGF-A) and C-C motif chemokine ligands (CCL) by astrocytes have been demonstrated to disarrange the blood-brain barrier and, therefore, cause tissue damage [129] and trigger infiltration by lymphocytes [130]. Furthermore, reactive astrocytes may affect, or be affected by, infiltrating immune cells. For example, Wheeler et al. [37] showed that reduced astrocyte expression of the Nuclear factor erythroid 2-related factor 2 (NRF2) was correlated with increased production of the small musculoaponeurotic fibrosarcoma G (MAFG) proteins that enhance the inflammatory responses of infiltrating T-cells in EAE mouse models. The same research group showed that astrocytes expressing high levels of lysosome-associated membrane protein 1 (LAMP-1) and the TNF-related apoptosis-inducing ligand (TRAIL) promoted apoptosis of infiltrating T-cells, thus performing an anti-inflammatory activity [131]. On the other hand, astrocyte expression of these markers was found to depend on their interactions with infiltrating NK cells, secreting the cytokine IFN-γ, and was inhibited by other cytokines such as tumor necrosis factor (TNF) and interleukin 1α (IL-1α), secreted by T cells. Contextual neuroprotective functions acquired by astrocytes in neuroinflammation may produce maladaptive effects in the long term. For example, under conditions of low glucose concentration astrocytes produce metabolites of fatty acids to provide energetic support to neurons, but also generate reactive oxygen species (ROS) [132]. It should be emphasized here that, except for this last study, in which 12–16 and 80 weeks old animal models of HD were used, all previous data were obtained from young adult mice (about 36 weeks old) or in vitro. This contextualization is necessary especially for animal models reproducing age-related diseases such as AD, PD and HD. The presence (or not) of reactive astrocytes in the aged brain is still a matter of debate [111]. However, although aging cannot be regarded as a disease, it is well accepted that young, adult and aged brains are not fully comparable organs. Therefore, it is conceivable that astrocytes modify their activities and show morphological changes in response to the dynamic modifications of the surrounding environment. Moreover, these functional changes of astrocytes may reversely contribute to altering the environment of the CNS and favoring the onset of age-related diseases.

## 5. Interactions between Microglia and Astrocytes

### 5.1. In Health

Microglia are the first detectable cells of neuroglia in the embryonic CNS, where they may play a role in the development of nerve tracts [133] as well as functional neuronal networks [134]. Evidence suggests that these cells are also involved in the generation of astrocytes from neuronal precursors during development: in mouse embryos, microglia gather close to neuronal progenitors [40], and the differentiation of astrocytes in primary cultures of mouse neuronal progenitors strictly depends on the presence of co-cultured microglia [135]. Furthermore, this inductive action appears to last in postnatal development, since microglia was found to promote astrocyte maturation in the hippocampus of newborn rats [136]. Conversely, in the hippocampus of both adult (12 weeks old) and aged (88 weeks old) rats, branching of microglia was found to be activated by the meshwork of astrocyte processes via dynamic cell–cell contacts [33] (Figure 1D–F). Eventually, the modulation of environmental conditions provides an additional strategy of mutual inductions, since both astrocytes and microglia mold the architecture and mechanical properties of the ECM. As stated above, constitutive activities of microglia and astrocytes overlap and concur to mediate remarkable functions in developing and mature CNS, such as modulation of synaptic connectivity and neurotransmission. Therefore, it is conceivable that tuning of overlapping functions may be performed through reciprocal interactions, along with orchestration by neurons. At the present time, there are few data available regarding these mutual influences. Indeed, suitable knowledge of how these constitutive heterotypic relationships may affect, or be affected by, the CNS environment in a lifetime course, could help to understand the eventual dysfunctional features shown by astrocytes and microglia in aging. Moreover, in the early phases of neuroinflammation, both astrocytes and microglia intensify their constitutive activities and gain new functions. Due to the increasing complexity of their involvement, tuning interactions between the two cell types should also be markedly intensified, which favors their dysregulation and promotes neurodegenerative effects. Therefore, a proper depiction of the interactions between glial cells in the healthy brain may provide the required background to understand the fast course of inflammation in neurodegenerative diseases.

### 5.2. In Neuroinflammation

The first detectable interaction between microglia and astrocytes in neuroinflammation concerns the induction of reactivity. About 30 years ago, Matsumoto and colleagues [137] found that “reactive microgliosis” preceded “reactive astrogliosis” in a mouse model of AD (8–12 weeks old). Based on this temporal sequence, it was assumed that microglia may induce astrocyte reactivity [1,138]. This relationship was then transposed into the A1/A2 paradigm, suggesting that M1 microglia may trigger the expression of the A1 phenotype of astrocytes [36]. However, in view of the marked dependence of astrocyte and microglia phenotypic heterogeneity from context conditions, this activation hierarchy seems to contrast the mutual interdependence linking the two cell types in CNS diseases. For example, recent evidence collected in vitro suggests a role of astroglia in the activation of microglial immune response induced by obesity [32]. To be recalled, the scenario of heterotypic interactions is even more complex and also involves neurons, which may promote microglial reactivity by releasing several signaling molecules such as the chemokines (C-C motif) ligand 2 (CCL2) [139] and fractalkine [140,141], and extracellular alarmin high mobility group box protein-1 (HMGB-1) [142], as well as by increasing extracellular levels of ATP [143] and glutamate [144]. Moreover, cytotoxic unfolded peptides on neurons and cell debris may also represent pro-inflammatory factors [7]. According to their neurotrophic and clearance roles, astrocytes are possibly able to rapidly sense the accumulation of these molecules in the early phases of neurodegenerative diseases, such as AD. Indeed, these cytotoxic molecules have been described to induce fragmentation of astrocyte processes (Figure 1G), resulting in disruption of astrocyte meshwork in aged rats [55]. Furthermore, structurally altered proteins activate the NF-κB signaling pathway [145] and inhibit the astrocyte support to synaptogenesis in prion-infected mice (3–4 weeks old) [146]. Whether or not these responses may be categorized as signs of astrocyte reactivity, they induce significant changes in the nervous tissue environment. In addition, the disruption of the astrocyte meshwork hampers heterotypic interactions in the aged nervous tissue. In particular, our study in the hippocampus of aged rats (88 weeks old) demonstrated that local interruptions of the astroglial meshwork imply a decrease in their direct interactions with microglia (Figure 1F) and are responsible for the microglial shift from branched to amoeboid morphology [33] (Figure 1C,F), thus providing a rationale of their impaired clearance efficiency. It appears that the impairment of typical astrocyte tasks addressed to nerve cells, such as trophic support and clearance activity may give rise to a context where the onset and progression of neurodegenerative diseases are favored. These data suggest that homeostasis alterations in the CNS may elicit multiple responses from different cell types, thus stressing the idea of neuroinflammation as a choral reaction to pathological stimuli.

It has been, however, assessed that in neuroinflammation, intense molecular crosstalk between glial cells is maintained via a variety of molecules: growth factors, gliotransmitters, cytokines, chemokines, innate-immunity mediators, ATP, mitogenic factors, nitric oxide (NO), ROS and glutamate. Recent evidence collected from a study on 17-week-old mice indicates that microglia promote neuroprotective response from astrocytes by releasing cytokines, such as interleukin-1 beta (IL-1β), Tumor Necrosis Factor alpha (TNF-α) and IL-6 [147]. On the other hand, reactive astrocytes may promote microglial phagocytosis by releasing the complement factor C3 [145] or ATP [148]. Moreover, astrocytes may either reverberate activation signaling to microglia, thus promoting their migration towards injury sites and phagocytosis, or inhibit their reactivity when physiological homeostasis is restored [1]. The exchange of extracellular vesicles containing active molecules, such as mRNA fragments is suggested as an additional mechanism of micro-astroglial interaction: in vitro, extracellular vesicles released by microglia, affect astrocyte expression of reactivity markers and the production of components of the extracellular matrix [149], whereas vesicles produced by astrocytes modulate microglial migration and phagocytosis [150]. Among these activities, astrocyte production of ECM molecules, such as fibronectin, represents a further strategy of modulating the microglial immune response. It is long known that such molecules may regulate microglial expression of integrin-β1 [151], a mechano-receptor involved in cell–ECM as well as cell–cell mechano-signaling [152,153]. A direct correlation between integrin-β1 expression and onset of reactivity in microglia was also demonstrated in cultures of murine primary cells [154] and ex vivo in mice models (8–10 weeks old) of MS [76]. A study carried out in 3D collagen substrata and organotypic slices assessed the dependence of microglia migration on their expression of integrin-β1 [155]. Our previous studies on direct cell–cell interactions between astrocytes and microglia indicated the recruitment of integrin-β1 at contact sites between their processes, along with focal increases of the concentration of ionized calcium-binding adapter molecule (Iba1) [33], a microglial marker of Ca^2+^ dependent cytoskeletal remodeling, involved in both migration and phagocytosis [20]. These data suggest that networks of mutual modulations, affecting inflammatory responses of astrocytes and microglia, may be utterly complicated by the integration of molecular and mechanical processes. A paradigmatic example of astrocyte-microglia interactions in neuroinflammation is provided by the occurrence of “triads”, in which astrocytes and microglia cluster with damaged nerve cells and promote clearance of cytotoxic neuronal debris from the nervous tissue [28,30,156]. In detail, processes originating from an astrocyte form a kind of “mini scar” encircling a terminally injured neuron, split the cell into debris and expose them to microglial phagocytosis.

It is known that long-term activation of the mutual inductions cycle linking microglial and astroglial reactivity may result in a decrease of their activities of maintenance and support, along with direct neurodegenerative effects. As previously stated, the high metabolic activity of reactive astrocytes and microglia may lead to the overproduction of ROS, which in turn, induces oxidative stress and neurodegeneration in the nervous tissue [12,132]. On the other hand, neurodegenerative effects of oxidative stress may be counterbalanced by the neuroprotective activity of NO [157]: in the hippocampus of adult rats, it has been demonstrated that astroglial production of NO induced the expression of heme-oxygenase-1, an enzyme involved in the synthesis of the known antioxidant bilirubin [158]. Moreover, the production of NO by microglia and astrocytes may trigger clearance of damaged cell debris and defense against invading bacteria [159]. Nonetheless, NO may inhibit axonal conduction and induce neurological disorders in adult rats [160] and enhance cytotoxicity mediated by NMDAR [12].

A high concentration of cytokines involved in astrocyte-microglia crosstalk during inflammation may also induce neurodegeneration. Evidence has been provided that injection of IL-1 in ischemic rats resulted in neurodegenerative effects [161]. Conversely, a study in mouse models of cerebral ischemia demonstrated that overexpression of IL-1 through viral transfection exerted beneficial effects [162]. Neurodegenerative induction by cytokines was also observed in transgenic mouse models overexpressing IL-6 (age: 12, 24 and 48 weeks) [163] and TNF-α (age: 11 weeks) [164]. Since these molecules are proinflammatory, it is not surprising that their neurodegenerative effects were found to involve other modulators of inflammation. This is the case of IL-1 [165] and TNF-α [166], which were demonstrated to activate NO production in co-cultured human astrocytes and neurons, whereas synergy between IL-6 and transforming growth factor β (TGFβ) generated pathogenic T_H_17 lymphocytes in in vitro and ex vivo experiments on mice [167]. Changes in the ECM may also be involved in the activation and, eventually, intensity regulation of the immune response. Evidence showed that expression of fibronectin by astrocytes may be increased following seizures induced by kainic acid (in 12-week-old rats) [168]. On the other hand, a study performed on animal models of spinal cord injury (12-week-old mice) indicated that increased expression of astroglial fibronectin may induce the exacerbation of the immune response [169]. These data seem to suggest that astrocyte and microglia during inflammation activate a cycle of self-triggering, mutual inductions, which coordinate but also steadily amplify their responses until either resolution of the disease or onset of new neurodegenerative processes. However, growing evidence indicates that non-inflammatory processes modulating synaptic connectivity are finely tuned by cytokines during development and in health. A representative example is provided by IL-33, a member of the IL-1 family that showed either neuroprotective or neurodegenerative effects in different experimental conditions. Intraperitoneal injections (50 mg/kg, about 1.15 mg per animal, 10 days) followed by intrahippocampal injection (400 ng by side) of IL-33 in 8-week-old mice, evoked neuroinflammation and cognitive impairment [170], whereas intraperitoneal injections (200 ng, 7 days) of IL-33 in transgenic mouse models of AD (mice, 48 weeks old) mobilized microglia to prevent and clear Aβ-deposits, thus ameliorating cognitive impairment [171]. Of note, IL-33 released by astrocytes has been suggested to modulate typical microglia activities such as the pruning of synapses in mouse embryos during maturation [50] and modulation of synaptic plasticity in adult mice (16 weeks old) [104]. In cell cultures, TNF-α has been demonstrated to promote astrogenesis from human neural progenitors [172] and the maturation of human neuroblasts [173]. These data indicate that cytokines may be involved, either as effectors or mediators, in dynamic lifespan changes of the CNS. In neuroinflammation, a massive increase of cytokine concentration causes a sudden, choral activation of neuroinflammatory processes and, at the same time, triggers inherent, mutually reinforcing counter effects. Indeed, a metabolic acceleration involves both astrocytes and microglia, promoting oxidative stress and, therefore, cell damage. The increasing cytotoxicity induced by these noxious counter effects is constantly weighted against neuroprotective effects of inflammation and, in the course of time, may result in a neurodegenerative outcome. Finally, in the aged CNS, a condition of chronic inflammation involving dysregulation of astrocyte-microglia interactions favors the onset of age-related neurodegenerative diseases.

## 6. Conclusions

The pattern of diverse interactions occurring between astrocyte-microglia in neuroinflammation has been the object of intense scientific investigation in recent years, due to its relevance in diffused and dramatic pathologies. The growing evidence of new molecular and mechanical modulatory processes, functionally linking these interactive cell types, had such a rapid onset that, in some cases, the main fascinating question was forgotten. Which are the mechanisms driving such diverse cell types to entertain such complex relationships? If we think of their different embryological origins it seems very unlikely that astrocytes and microglia could establish such a wide and variegate range of interactions, and it is even more difficult to consider their interactions with other cell types in the nervous tissue such as neurons, oligodendrocytes, endothelial cells of blood vessels and so on. Indeed, data here reviewed suggest that these interactions are remarkable components of—and concur to shape—the environment of the CNS in the lifespan, thus modulating broad spectra of different responses and/or phenotypes of all cell types. Even the patterns of distribution of the different cell types, during embryogenesis as well as in postnatal maturation and adulthood, apparently facilitate reciprocal interactions.

Although some general statements have been proposed in the past, as for the microglia-astrocyte hierarchy in the induction of reactivity, the identification of a reliable chain reaction linking the responses of the different cell types in the different pathologies still appears an extremely difficult, almost impossible, task.

It remains unchanged the necessity to further detail the links and nodes of this network of mutual interdependencies as they concur to depict a functional architecture within the whole CNS. On the other hand, it is also mandatory to stimulate integrated studies, in which the theoretical background results from the melting of physical, molecular, cellular and histological data at the “whole system” scale.

Recent technological advances enabling the collection and management of large amounts of data per se justify the necessity of a multi-scaled synthesis of the current knowledge on the CNS to understand how it reacts “as a whole”, to the different lifespan challenges.

## Figures and Tables

**Figure 1 cells-10-01195-f001:**
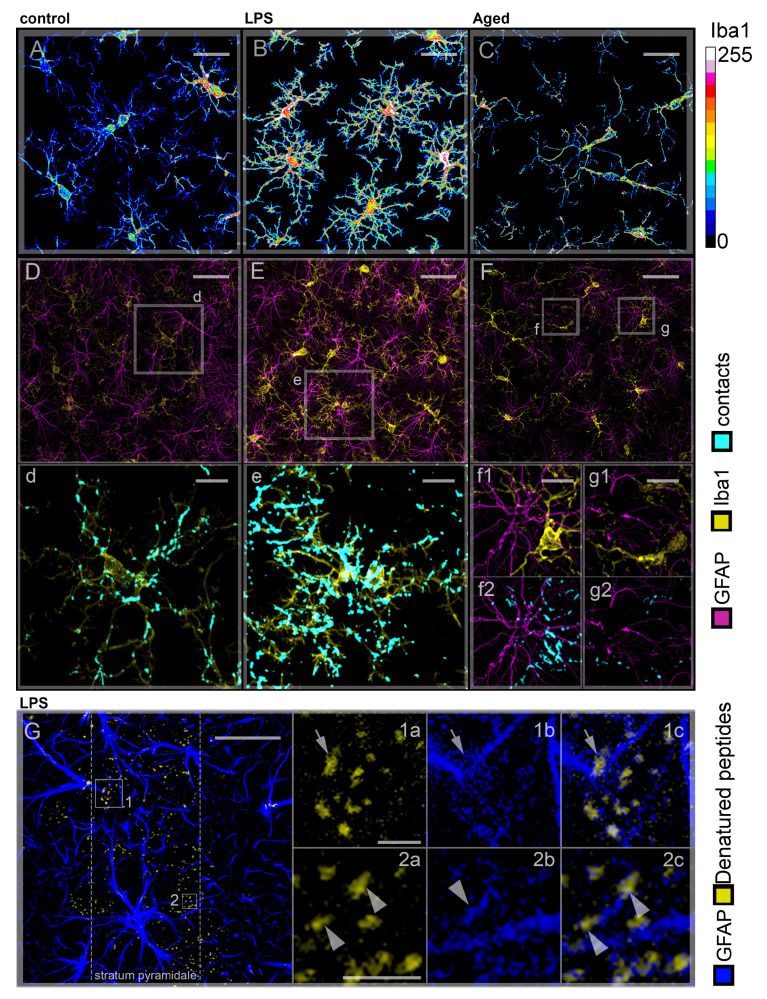
(**A**–**C**) Ca^2+^ dependent cytoskeletal remodeling is involved in microglial branching. Microglia in the CA1 hippocampus of adult control rats (**A**), adult rats treated with lipopolysaccharide (LPS) infusion (1.6 mg/mL LPS, 4 weeks) to induce neuroinflammation (**B**), and aged rats (**C**) were stained to reveal Iba1 (16-color LUT), a marker of cytoskeletal remodeling. In LPS-treated rats (**B**) microglia show increased branching and expression of Iba1 in comparison with control rats (**A**). In aged rats (**C**), microglia display elongated soma and limited branching [33]. (**D**–**F**) Microglial branching and density of the meshwork formed by astrocyte processes are correlated. Microglia and astrocytes in the CA1 hippocampus of adult control (**D**), adult LPS-treated (**E**), and aged (**F**) rats were immunostained to reveal Iba1 (yellow LUT) and GFAP (magenta LUT). Insets d–g) Details of the areas selected in (**D**) (inset d), (**E**) (inset e) and (**F**) (insets f and g), showing cell–cell contacts (cyan LUT). Increased branching and Iba1 expression of microglia in LPS—treated with respect to control—rats appear directly correlated with a proportional increase of astrocyte meshwork density (**D**,**E**) and cell–cell contacts (insets d,e). The same correlation was found in aged rats (**F**): branched microglia (inset f1) were found within intact astrocyte meshwork establishing numerous cell—cell contacts (inset f2), whereas amoeboid unbranched microglia (inset g1) were found in areas showing meshwork disruption (inset g2) [33]. **G**) Contacts between astrocytes and autofluorescent deposits of denatured peptides on neurons promote process fragmentation. Sections of the CA1 hippocampus of LPS-treated rats were immunostained to reveal GFAP (blue LUT); autofluorescence of denatured peptides was also collected (yellow LUT). Insets 1 and 2, details of the areas selected in (**G**). Astrocyte processes diffusely contact denatured peptides produced by the neurons of the stratum pyramidale (dashed lines). Arrows in inset 1 indicate a deposit (1a) of denatured peptides on a fracturing astrocyte process. Arrowheads in inset 2 indicate two peptide deposits associated with a GFAP+ fragment [55]. Scale bars: A–C, G = 25 µm; D–F = 40 µm; Insets d, e = 10 µm; Insets f, g = 5 µm; Insets 1, 2 = 3 µm.

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
