# Peer review of "Neuroinflammation: Integrated Nervous Tissue Response through Intercellular Interactions at the “Whole System” Scale"

_cells, 2021, doi:10.3390/cells10051195_

Round 1

Reviewer 1 Report

In this review manuscript (cells-1194294), the authors discussed neuroinflammation mediated by microglia and astrocytes as a whole system. This topic is interesting. Some concerns and suggestions are listed as below:

  1. The major concern is regarding M1 and M2. In fact, the adoption of M1 and M2 was undertaken in an attempt to simplify data interpretation at a time when the ontogeny and functional significance of microglia had not yet been characterized. Now, terminology suggesting established meaningful pathways of microglial polarization hinders rather than aids research progress and should be discarded (Ref: A polarizing question: do M1 and M2 microglia exist? Nature Neuroscience, 2016).
  2. Another concern is regarding A1 and A2. This is not well accepted in glial field. I would suggest that the authors use MAFG+ astrocytes and LAMP+TRAIL+ astrocytes instead. Please read two recent Nature papers from Quintana’ lab at Harvard Medical School.
  3. Please change the description ‘cell line’ throughout the manuscript.
  4. Disease-associated microglia, a recently identified subset of CNS resident macrophages found at sites of neurodegeneration, might play a protective role. This point should be discussed.
  5. The authors provided a description of astroglia-microglia interactions during inflammation, with the aim of discussing how they might affect, or be affected by the whole system. How about the role of neurons in the whole system?
  6. The authors said that microglia differ from peripheral macrophages in terms of protein expression and adaptive potential to different environmental conditions. Please discuss this point in details.
  7. Emerging data has convincingly demonstrated the existence of sex-dependent structural and functional differences of microglia (Uncovering sex differences of rodent microglia, Journal of Neuroinflammation, 2021). Sex should be considered as a biological variable when designing microglial experiments. This point should be discussed in the revised manuscript.
  8. The authors said that under physiological conditions microglia are localized in the gray matter. However, microglia are also located in white matter. Potential differences of microglia in white matter and gray matter (brain-region specific effects) should be discussed.
  9. In the part of microglia in health, the authors did not need to talk about aging and AD in this section.
  10. In the part of ‘Microglia in Neuroinflammation’, it is not clear for readers which neuroinflammatory conditions were mentioned. Apart from aging, multiple sclerosis and its animal model should also be discussed in this section.
  11. In figure 1, scale bars should be added.
  12. Apart from Iba1 staining in figure 1, novel microglial specific markers such as Tmem119 and P2ry12 should be used. How about the quantification of infiltrating macrophages after LPS injection?
  13. In figure 1, the dose of LPS should be mentioned.

Reviewer 2 Report

The present manuscript is well-structured, well-written and easy to understand.

The title is Neuroinflammation: Integrated Nervous Tissue Response through Intercellular Interactions at the “Whole System” Scale. So, Microglia in Neuroinflammation is one of the key cores of this review.

Neuroinflammation is initiated by microglia, which are the resident immune cells of the central nervous system. Under steady-state conditions, microglia are maintained in a “resting” state through interactions with cell surface and soluble factors from surrounding cells.

As the author mentioned, Currently, we know that the transition of microglia activity from neuroprotective to neurodegenerative is dependent upon time. In fact, when physiological homeostasis of the CNS is quickly restored, microglia return to their “resting” state.

Mitochondria are classically known to be cellular energy producers. Given the high-energy demanding nature of neurons in the brain, it is essential that the mitochondrial pool remains healthy and provides a continuous and efficient supply of energy. However, mitochondrial dysfunction is inevitable in aging and neurodegenerative diseases.

These protein inhibitors are involved in quickly tamping down the inflammatory response after spinal cord injury, The microglia essentially orchestrated swift removal of harmful cell debris after injury and stopped inflammation.

Many related updated publications not cited in this review, such as below.

Microglia-organized scar-free spinal cord repair in neonatal mice Nature. 2020 Nov;587(7835):613-618.

A Glimmer of Hope: Maintain Mitochondrial Homeostasis to Mitigate Alzheimer’s Disease Aging and disease    2020, Vol. 11  Issue (5) : 1260-1275     DOI: 10.14336/AD.2020.0105

Mitochondria, Microglia, and the Immune System—How Are They Linked in Affective Disorders?

Front. Psychiatry, 09 January 2019 | https://doi.org/10.3389/fpsyt.2018.00739

Microglial mitophagy mitigates neuroinflammation in Alzheimer's disease

Neurochemistry International Volume 129, October 2019, 104469

I strong suggest the author list the Microglia and Mitochondria in the section of Microglia in Neuroinflammation.

Round 2

Reviewer 1 Report

The authors have addressed my concerns.